# Circulating Tumor Cells and Bevacizumab Pharmacokinetics during Neoadjuvant Treatment Combining Chemotherapy and Bevacizumab for Early Breast Cancer: Ancillary Analysis of the AVASTEM Trial

**DOI:** 10.3390/cancers13010140

**Published:** 2021-01-05

**Authors:** Renaud Sabatier, Jean-Yves Pierga, Hervé Curé, Rakan Abulnaja, Eric Lambaudie, François-Clément Bidard, Jean-Marc Extra, Patrick Sfumato, Anthony Gonçalves

**Affiliations:** 1CNRS U7258, INSERM U1068, Institut Paoli-Calmettes, CRCM, Aix Marseille University, 13009 Marseille, France; lambaudiee@ipc.unicancer.fr (E.L.); GONCALVESA@ipc.unicancer.fr (A.G.); 2Department of Medical Oncology, Institut Paoli-Calmettes, 13009 Marseille, France; dr.rabulnaja@gmail.com (R.A.); extrajm@ipc.unicancer.fr (J.-M.E.); 3Department of Medical Oncology, Institut Curie, Paris & St Cloud, Université de Paris, 75005 Paris, France; jean-yves.pierga@curie.fr (J.-Y.P.); francois-clement.bidard@curie.fr (F.-C.B.); 4Department of Medical Oncology, Institut Jean Godinot, 51100 Reims, France; hcure@chu-grenoble.fr; 5Department of Surgical Oncology, Institut Paoli-Calmettes, 13009 Marseille, France; 6Department of Clinical Research and Innovation, Biostatistics Unit, Institut Paoli-Calmettes, 13009 Marseille, France; sfumatop@ipc.unicancer.fr

**Keywords:** early breast cancer, bevacizumab, neoadjuvant chemotherapy, circulating tumor cells and pharmacokinetics

## Abstract

**Simple Summary:**

We recently published the results of the AVASTEM study, in which we explored the impact of the addition of an angiogenesis inhibitor (bevacizumab) to standard pre-operative chemotherapy for breast cancer. In this work, we aimed to identify biological parameters correlated to prognosis and treatment efficacy. To do so we explored if circulating tumor cells (CTCs), that are cells released by the tumor and that can be detected in the bloodstream of a patient; can predict the outcome of the women treated in this study. We also analyzed if bevacizumab concentration in the blood during treatment was associated with outcome, i.e., response to treatment and survival. We observed that CTCs detection before treatment initiation is associated with survival but was not correlated to local response to treatment. We moreover found that CTC status after three weeks of chemotherapy was not correlated to outcome as well as bevacizumab levels before and during treatment. However, bevacizumab concentration tended to be associated with an increase of hematological toxicities during the study. We thus show in this work that CTCs detection at baseline is prognostic for patients with breast cancer receiving a pre-operative chemotherapy-bevacizumab combination, and that this prognostic value was independent of tumor response.

**Abstract:**

The phase II AVASTEM trial explored the impact of chemotherapy-bevacizumab combination on breast cancer stem cells in the neoadjuvant setting. We aimed to identify biological features associated with preoperative chemotherapy efficacy and prognosis by analyses of circulating tumor cells (CTCs) and bevacizumab pharmacokinetics (PK). The main objective was to assess the prognostic (relapse-free survival and overall survival) and predictive (pathological complete response, pCR) values of CTCs (CellSearch technology) and bevacizumab PK (ELISA). Seventy-five patients were included. Out of them 50 received bevacizumab-chemotherapy and 25 received chemotherapy alone. CTC results were available for 60 patients and PK data for 29 patients in the experimental arm. The absence of CTC at inclusion was correlated to better outcome. Five-years overall survival (OS) was 91% for CTC-negative patients vs. 54% for CTC-positive cases (HR = 6.21; 95%CI (1.75–22.06), *p* = 0.001, log-rank test). Similar results were observed for RFS with 5 y-RFS of 78% vs. 44% (HR = 3.51; 95%CI (1.17–10.52), *p* = 0.017, log-rank test). However, CTC status at baseline was not predictive of pCR (*p* = 0.74). CTC status after one cycle was not a significant prognostic factor (HR = 1.56; 95%CI (0.19–12.67); *p* = 0.68 for OS and HR = 2.76; 95%CI (0.60–12.61); *p* = 0.17 for RFS, log-rank test). Bevacizumab serum levels could not predict pCR and survival. PK values were not associated with treatment-related toxicities. In conclusion, CTCs detection at baseline is a prognostic marker for breast cancer receiving a neoadjuvant chemotherapy-bevacizumab combination independently of tumor response.

## 1. Introduction

Breast cancer is a public health issue. According to the ECIS (European Cancer Information System, https://ecis.jrc.ec.europa.eu/) in 2020, the predicted number of new breast cancers in 27 European Union countries is 355,457, with estimated mortality of 34.1/100,000, with 91,826 predicted deaths.

Treatment of stage II and III early breast cancer is based on combinations of surgery, radiation therapy, endocrine therapy for estrogen receptor positive cases, trastuzumab for HER2 (human epidermal growth factor receptor 2)-positive tumors, and cytotoxic chemotherapy. Neoadjuvant chemotherapy (systemic chemotherapy before breast and lymph nodes surgery) aims to downstage the tumor to enable breast-conserving surgery, evaluate the effectiveness of new systemic therapy or new therapeutic schemes, and to eradicate micro metastases and to reduce the risk of distant recurrence [1,2]. Most regimens used in this setting are sequential combinations of anthracyclines and taxanes with addition of trastuzumab for HER2-positive tumors.

Addition of bevacizumab to these regimens has been shown to increase pathological response rates [3] but there is no evidence that it can improve survival in this setting. As no relevant circulating biomarker has been described to be predictive of bevacizumab efficacy [4], identification of better predictive markers may help clinicians to improve patients’ selection as well as the efficacy of targeted therapies.

Circulating tumor cells (CTC) are released by the main tumor mass and may be at the origin of metastases spreading [5]. They have been shown to be correlated to survival and to prediction of chemotherapy efficacy in the metastatic setting [6]. CTC can be identified for 2 to 55% of patients with primary breast cancer, with a detection rate below 25% with the CellSearch^®^ system which is the only assay that has received FDA approval [7]. CTC detection in patients with early breast cancer is associated with tumor size, lymph node involvement, and high grade [8,9]. CTC detection at diagnosis is also correlated to a poorer outcome with decreased disease-free survival and overall survival in chemonaive patients diagnosed with early breast cancer [10,11]. Moreover, CTC detection after neoadjuvant chemotherapy is associated with worst survival for early triple-negative breast cancer with residual disease after preoperative chemotherapy [12]. No data exist concerning CTC value to assess bevacizumab efficacy for early non-inflammatory breast cancer.

Despite bevacizumab pharmacokinetics characteristics have been showed to be associated with treatment efficacy in the metastatic setting [13], its impact on neoadjuvant chemotherapy efficacy remains unclear.

We previously published the main results of the AVASTEM trial (NCT01190345), which showed that addition of bevacizumab did not modify the stem cells rates in comparison to standard neoadjuvant chemotherapy [14]. The current work is based on CTC and PK ancillary analyses of blood samples collected during this trial. We explored the association between CTC detection, CTC kinetics during treatment, bevacizumab blood concentration, and treatment efficacy (pCR achievement and survival). PK was also compared to the incidence of selected treatment-related adverse events.

## 2. Results

### 2.1. Patients’ Clinicopathological Features

Among the patients included in the AVASTEM trial, treatment groups were well balanced in terms of age, menopausal status, tumor stage, and axillary lymph node involvement [14].

We obtained CTCs data at inclusion for 60 patients, including 38 who received bevacizumab and 22 who received chemotherapy alone. Among them, percentages of positive axillary lymph node and De novo metastatic patients were significantly different between CTC positive and negative subgroups (100% vs. 75% *p* = 0.05 and 33% vs. 7% *p* = 0.02, respectively, Table 1). CTCs data were still available after one cycle of treatment (post-C1 time point) for 56 patients (35 in the experimental arm and 21 in the control arm) with no significant differences between subgroups for demographic and clinical factors.

PK values were available at least one time point for 38 patients (29 in the experimental arm and 9 in the control arm): twenty five patients at inclusion (19 in the experimental arm and 6 in the control arm), 27 after cycles 1–2 (22 in the experimental arm and 5 in the control arm), and for 22 after end of treatment (18 in the experimental arm and 4 in the control arm).

Demographic and clinical factors for the 60 patients with at least one CTC measure and the 29 patients in experimental arm with at least one PK measure are also presented in Appendix A.

### 2.2. CTC Analyzes

At inclusion CTCs were identified for 15 of 60 patients (25%; nine received bevacizumab and 6 had been randomized in the control arm, *p* = 0.77). After one cycle (sample collected at C2D1), CTCs were still observed in five of 56 patients (8.9%; two of them were in the bevacizumab arm and three were in the control arm, *p* = 0.35, Table 2).

CTC positivity at baseline was not correlated to pCR achievement (Table 3). Response was evaluated for 56/60 patients who had CTCs data at baseline (16 with pCR and 40 with RD) and for 54/56 patients with CTCs data after one cycle of treatment (15 with pCR and 39 with RD). At inclusion, three of the 16 patients with pCR had detectable CTCs versus 10 of the 40 with RD (18.75 vs. 25%, *p* = 0.74). After one cycle of treatment, one of the 15 patients with pCR status had still detectable CTCs versus four of the 39 with RD (6.7 vs. 10.3%, *p* = 1).

The absence of CTC at inclusion was associated with better outcome. Figure 1A shows that 5 y-OS was statistically higher in patients without CTC at baseline (91% (95%CI (77–96))) compared to CTC-positive patients (54% (95%CI (25–76))), (Hazard ratio = 6.21 (95%CI (1.75–22.06); *p* = 0.001, log-rank test). This survival gain was not observed when analyzing CTC status after one cycle of treatment. Figure 1B shows that 5 y-OS was similar for patients without post-C1 CTC (86% (95%CI (72–93))) and for patients who were still CTC-positive at C2D1 (80% (95%CI (20–97))), (HR = 1.56; 95%CI (0.19–12.67); *p* = 0.68, log-rank test).

Regarding non-De Novo metastatic patients, Figure 2A shows that the absence of CTC at inclusion was associated with significantly better relapse-free survival with a 5 y-RFS rate of 78% (95%CI (61–88)) versus 44% (95%CI (14–72)) for CTC-positive patients (HR = 3.51; 95%CI (1.17–10.52), *p* = 0.017, log-rank test). CTC status after one cycle was not a significant prognostic factor in our set. Figure 2B shows that the 5 y-RFS rate was 77% (95%CI: (61–87)) for post-C1 CTC-negative patients versus 50% (95%CI: (6–84)) for post-C1 CTC-positive cases (HR = 2.76; 95%CI (0.60–12.61); *p* = 0.17, log-rank test).

### 2.3. PK Analyzes

Bevacizumab measurements were similar at baseline for both treatment cohorts: 3.8 µg/mL vs. 2.25 µg/mL, *p* = 0.426. As expected, bevacizumab levels were higher in the experimental arm after treatment initiation and after end of treatment, with a median of 111.2 µg/mL (range 43.0–640.9) and 74.2 µg/mL (1.0–956.5), respectively (Figure 3).

When looking at post C1/C2 PK correlation to pCR in the experimental arm, we observed that bevacizumab concentration did not correlate to pathological complete response achievement. Background baseline bevacizumab levels were similar whatever response to treatment was median of 3.3 µg/mL for patients with pCR versus 2 µg/mL for patients with RD (*p* = 0.421, Figure 4). After one or two cycles the median of PK concentration in case of pCR was 98.5 µg/mL versus 166.2 µg/mL for patients with RD (*p* = 0.664, Figure 4). After 8 cycles, the median for PK concentration were not significantly different for patient with pCR with 84.40 vs. 64 µg/mL in case of RD (*p* = 0.155, Figure 4).

Measurements of bevacizumab were not correlated to either OS (*p* = 0.86 at baseline and *p* = 0.122 after the end of treatment) neither RFS (*p* = 0.617 at baseline and *p* = 0.291 after the end of treatment).

### 2.4. Correlation of PK to Bevacizumab-Related Adverse Events

Anemia was notified for 13 of 50 patients (26%) in the bevacizumab arm. Seventeen patients (34%) had been admitted with febrile neutropenia (FN), and 14 (28%) were diagnosed with bevacizumab-induced hypertension (BIH). Bevacizumab levels after one to two cycles were not correlated to occurrence of BIH (*p* = 0.895) but tent to be correlated to hematological adverse events (*p* = 0.065 for FN and *p* = 0.084 for anemia), (Figure 5).

## 3. Discussion

In this retrospective ancillary analysis of the randomized phase II AVASTEM trial, we aimed to find biological markers that may help to predict efficacy and response to neoadjuvant chemotherapy in breast cancer.

We found that the absence of CTCs before treatment initiation was a prognostic factor associated with better OS (5 y-OS of 91% versus 54%, *p* = 0.001, log-rank test) and RFS (5 y-RFS of 78% versus 44%, *p* = 0.017, log-rank test). However, CTC status after treatment initiation (after one cycle) was not a prognostic factor in our set (*p* = 0.68 for OS and *p* = 0.17 for RFS). The low number of patients who were still CTC-positive after the first cycle of treatment may be an explanation of this absence of correlation. However, we cannot rule out that CTC measurement after one cycle of therapy is not an adequate surrogate marker of long-term efficacy and may not affect survival. Moreover, treatment modality (here bevacizumab vs. no bevacizumab) may also have modified the prognostic value of CTC status. It has been recently shown that treatment modalities can indeed influence the prognostic impact of clinical and biological features [15].

CTC has been described for the time for cancer during the 19th century [16]. Since this first publication, and notably in the last three decades, several data have been collected both in the early setting and for metastatic breast cancer cases [17,18]. In the metastatic setting, a large meta-analysis including nearly 2000 patients showed that both CTC count at baseline and one to two months after treatment initiation were associated with OS and progression-free survival, with a better accuracy compared to usual serum markers such as ACE and CA15-3 [19]. This was concordant with the first large dataset published ten years before [20]. Similar results were published for specific subtypes, including triple-negative breast cancer [21]. Concerning early breast cancer, CTC detection has been very consistent across studies, with rates ranging from 11 to 23%, and with higher rates for inflammatory breast tumors [22,23,24,25]. We observed a similar result in our study, with 25% (15/60) of CTC-positive cases at baseline, confirming the validity of our observations.

Despite its capacity to anticipate prognosis of patients with early breast cancer, CTC count predictive value is still not proven. Pathological complete response was not correlated neither to the status of CTCs nor to the PK values in our set. CTC-positivity rate at baseline was similar for patients who achieved pCR and for those with RD (18.75% for pCR vs. 25% for RD, *p* = 0.74). CTC status after one cycle of chemotherapy was also not predictive of response (6.67% for pCR vs. 10.26% for RD, *p* = 1). The lack of significant predictive value of CTCs has already be described for patients receiving neoadjuvant treatments combining chemotherapy and HER2-inhibitors (trastuzumab and lapatinib) in the NeoALLTO trial [23]. In this trial 27.3% of patients with detectable CTC displayed pCR, compared to 42.5% without CTC. A large meta-analysis that enrolled more than 1500 patients with documented baseline CTC-status also did not find any correlation between CTC detection and pCR achievement [26]. Taking this into account plus the prognostic value of CTC suggests that CTC-status may be more correlated to distant relapse and cannot predict sensitivity to systemic therapies. However, closer monitoring of CTC count during neoadjuvant treatment as well as molecular characterization of CTC may allow to improve their predictive value and lead clinicians to propose more intensive treatments to patients who remain CTC-positive after chemotherapy initiation [27].

For patients treated in the experimental arm, and who thus received an anthracycline-taxanes-bevacizumab combination, pharmacokinetics of bevacizumab was not associated with outcome. We observed a significant increase in bevacizumab serum concentration for patients enrolled in the experimental arm as soon as a few weeks after treatment initiation. This is consistent with the known half-life (~19 days) and previously published PK data [28]. Average blood concentration of bevacizumab has indeed been described to be 88 mg/L after the first dose [29]. These data are very close to ours (median of 111 mg/L after one or two cycles) suggesting the robustness of our results. Moreover, blood concentrations after the eight cycle were similar to those collected a few weeks after treatment initiation, in line with the hypothesis of a steady state after 100 days of treatment [30]. However, bevacizumab blood concentration was correlated neither to pCR achievement nor to long-term survival benefit of bevacizumab in our set. This lack of predictive or prognostic value of PK data is not limited to bevacizumab. PK features of other monoclonal antibodies used in oncology, and more specifically in breast cancer have also failed to predict outcome. PK analyses from the APHINITY trial demonstrated that pertuzumab exposure was similar in patients without relapse and in those with disease-free survival events [31]. Impact of PK may be different for antibody-drug conjugate as chemotherapy concentration is correlated to efficacy. Concerning advanced breast cancer, trastuzumab-emtansine exposure tent to be correlated to survival in patients included in the TH3RESA study [32].

Another hypothesis we addressed was that PK may be correlated to toxicity. We focused our analysis on three major adverse events under chemotherapy-bevacizumab regimen: anemia, febrile neutropenia, and hypertension. A quarter to a third of the patients included in our set experimented each of these toxicities (anemia 26%, FN 34%, and BIH 28%). We observed no correlation between bevacizumab blood concentration after treatment initiation and BIH. Hematological toxicities (FN and anemia) tent to occur more frequently in patients with high bevacizumab exposure (*p* = 0.065 and *p* = 0.086, respectively). Bevacizumab has been described to enhance hematological toxicities. Grade three or higher hematological adverse events (and specially FN) were indeed more frequent in the bevacizumab arm of the AVASTEM study, as well of other randomized trials [14,33,34,35,36]. However, this is, to our knowledge, the first time that the correlation between bevacizumab PK and adverse events is described in a set of patients receiving bevacizumab. Nevertheless, some scarce data related to the impact of blood exposure of other monoclonal antibodies have been presented. PK of HER2-inbibiting monoclonal antibodies (trastuzumab and pertuzumab) do not influence treatment-related toxicities [37,38].

One of the limitations of this study is that it is a retrospective analysis with limited number of patients. This precludes us from performing more extensive multivariate analyses that could have confirmed previously published data suggesting that CTC prognostic value is independent from usual clinicopathological features. More CTC timepoints could also have improved our analysis of correlation between CTC kinetics and response to systemic therapies. Moreover, more comprehensive PK data (Cmin, Cmax, area under the curve at steady state, clearance) would have been of interest to improve the understanding of relationships between drug exposure and efficacy or safety; and correlation between PK and hematological adverse events would probably be significant with more patients included in these analyses. However, as this PK/efficacy comparison was a retrospective ancillary of the AVASTEM trial, we were not able to collect more samples.

## 4. Materials and Methods

This study is a retrospective ancillary analysis of an open-label, randomized (2:1), multicenter, phase II trial with chemotherapy with or without bevacizumab combination for the treatment of primary breast cancer candidate to receive preoperative chemotherapy. Its primary objective is to evaluate the effect of circulating tumor cells (CTC) and pharmacokinetics (PK) of bevacizumab on pathological complete response rates (pCR). Secondary objectives are correlation of CTC and kinetics at baseline and after one cycle of treatment on relapse-free survival (RFS) and overall survival (OS). As an exploratory analysis, PK data were compared to incidence of the main adverse events observed with chemotherapy and bevacizumab combination: anemia, febrile neutropenia, and high blood pressure.

### 4.1. Patients

Blood samples available from the 75 patients included in the AVASTEM randomized controlled trial (EudraCT Number: 2009-014773-40; NCT01190345—August 2010) were analyzed. Inclusion criteria and treatments received have been previously detailed [14]. Fifty patients received standard neoadjuvant treatment plus bevacizumab; the other 25 received standard treatments only.

### 4.2. Blood Samples Collection Timeframe

Collection of CTC was done at baseline and after one cycle of treatment. For the PK, sample collection was done at baseline, after one or two cycles of chemotherapy, and after the eighth cycle.

### 4.3. CTC Analysis

CTC were detected from 7.5 mL blood samples using the Cellsearch^®^ system (Maniri Silicon Biosystems, Bologna, Italy) within 96 h by experienced technicians as previously described [39]. The CTC were measured before the beginning of the chemotherapy as either absence or presence. We added a second time point after one cycle to identify CTC-positive cases at baseline who were CTC-negative after one cycle of treatment.

### 4.4. Bevacizumab PK Analysis

Bevacizumab was provided by Abreos Biosciences Inc. and antibodies were obtained from the UCSD Moores Cancer Center Pharmacy.

All peptides were synthesized as C-terminal free acids and N-terminal free amines with an additional C-terminal GGG-Lys(biotin) for assay immobilization (Genscript, Piscataway, NJ, USA; CPC Scientific, Sunnyvale, CA, USA, or Abbiotec, San Diego, CA, USA).

ELISA Protocol. All incubations were conducted at room temperature on an orbital shaker unless otherwise noted. All washes were performed manually.

Wells of a NeutrAvidin-coated 96-well plate or 8-well strips (Thermo Fisher, Waltham, MA, USA) were coated with 100 µL of biotinylated peptide at 1 µg/mL in Hyclone H_2_O for 1 h. Wells were washed five times with 200 µL TBST. Wells were then blocked with 200 µL of 5% BSA in TBST for 1 h to prevent nonspecific binding. Wells were washed five times with TBST. Clinical Samples were diluted in 2.5% BSA/TBST to a final concentration of 0.2% and incubated on the plate for 1 h. Nonselective serum component were washed away 5 times with TBST. Bound bevacizumab was detected with goat anti-human Fc IgG-HRP (Jackson Immunoresearch, West Grove, PA, USA) diluted 1:5000 in BSA/TBST. Wells were incubated with 100 µL of diluted anti-IgG-HRP for 30 min and then washed nine times with TBST. Ultra TMB substrate (Thermo Fisher) was added to the wells (100 µL) and allowed to develop for 1–2 min shielded from light. The reaction was stopped with 100 µL of 1 M H_2_SO_4_ (Sigma, St. Louis, MO, USA) Optical density (OD) was read at 450 nm on a EnVision Xcite Multilabel S/N (Perkin Elmer, Waltham, MA, USA) and data were analyzed with GraphPad Prism v5 or v7 software (Graphpad Software Inc., La Jolla, CA, USA).

Each sample was run in duplicate and the resulting OD values were averaged. Reported bevacizumab concentrations were interpolated from the averaged ODs using a 4PL fit to a calibrator dilution series run on each plate. Calibrators used for this assay were made by spiking concentrations of bevacizumab into 0.2% commercial pooled human serum. These nominal concentrations were 500, 250, 125, 62.5, 31.25, 15.625, 7.8, 3.9, and 0 ng/mL.

Quality Control (QC) samples were included in the assays performed to confirm accuracy of concentration measurements. These samples were made by spiking bevacizumab into commercial 0.2% pooled human serum, as with the calibrators. The nominal spiked concentrations were 100, 50, and 20 ng/mL.

### 4.5. Statistical Analyzes

Categorical variables (treatment groups, menopausal status, tumor stage, axillary lymph node involvement, treatment response, and CTC status) were described using counts and frequencies, and quantitative variables (age and PK values) were described using medians and ranges.

Pathological complete response (pCR) was based on the ypT0/is pN0 definition. Patients without pCR or with disease progression during neoadjuvant treatment were considered as patients with residual disease (RD). CTC levels were compared to response and treatment arm using Fisher’s exact test. PK values were compared according to treatment arms and, for bevacizumab arm only, according to adverse events occurrence and response using the Wilcoxon rank sum test.

Survival analyzes were conducted on all patients for OS and on non-De Novo metastatic (M0) patients for RFS. Survival probability of outcomes was considered between the measurement of interest variable (C1D1 for baseline CTC and PK, C2D1 for post-C1 CTC status, C3D1 for post-C1/C2 PK, and surgery date for post-C8 PK) and the occurrence of interest event: death from any cause for OS and disease recurrence or death for RFS. Patients without any event were censored at the date of last contact. Survival curves were estimated using the Kaplan–Meier method and compared using the Log-rank test. Hazard ratios were provided with their bilateral 95% confidence interval (95%CI) and Wald’s test for significance. All tests were two-sided. The level of statistical significance was set at α = 0.05. Statistical analyses were carried out with the SAS^®^ software version 9.4 (SAS Institute Inc., Cary, NC, USA).

## 5. Conclusions

CTC detection and bevacizumab had no correlation to pCR for breast cancer patients included in the AVASTEM trial. The absence of CTCs at inclusion was associated with improved survival. Bevacizumab concentrations under treatment or after treatment completion were not correlated to survival and treatment-related adverse events. Larger prospective data are warranted to implement CTC use in routine practice for prognosis assessment of patients with breasts cancer treated in the neoadjuvant setting.

## Figures and Tables

**Figure 1 cancers-13-00140-f001:**
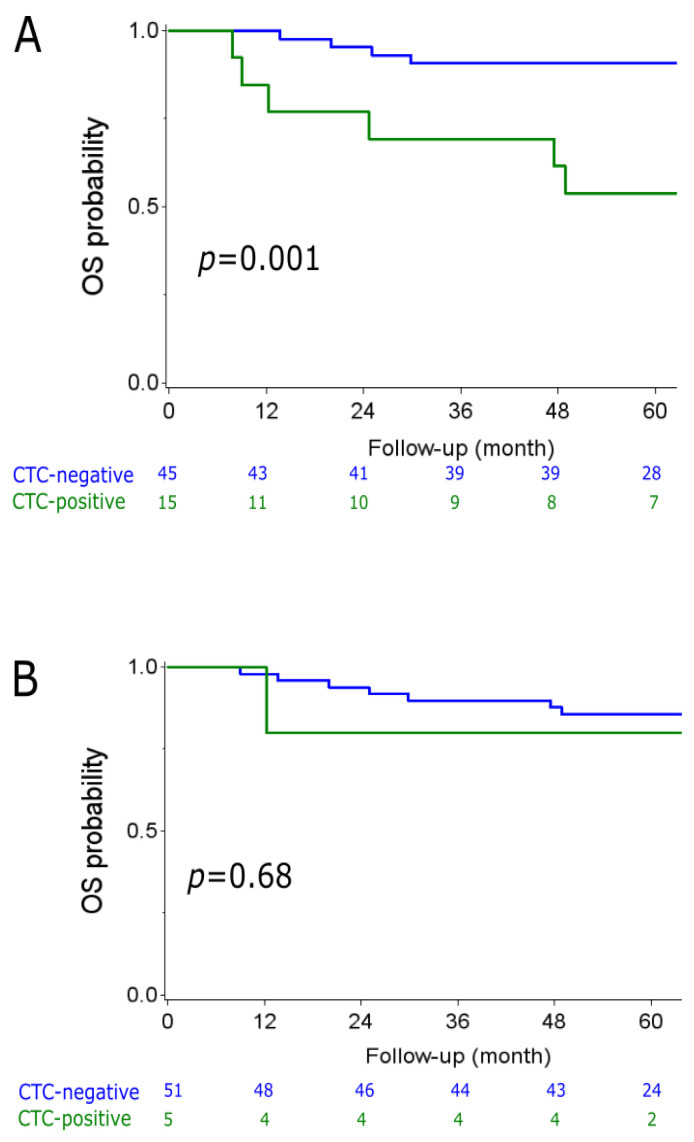
Kaplan–Meier curves for overall survival (OS) depending on the CTC status (green = CTC-positive; blue = CTC-negative). (**A**): CTC status at baseline. (**B**) CTC status after one cycle of treatment. *p* = Log-rank *p*-value.

**Figure 2 cancers-13-00140-f002:**
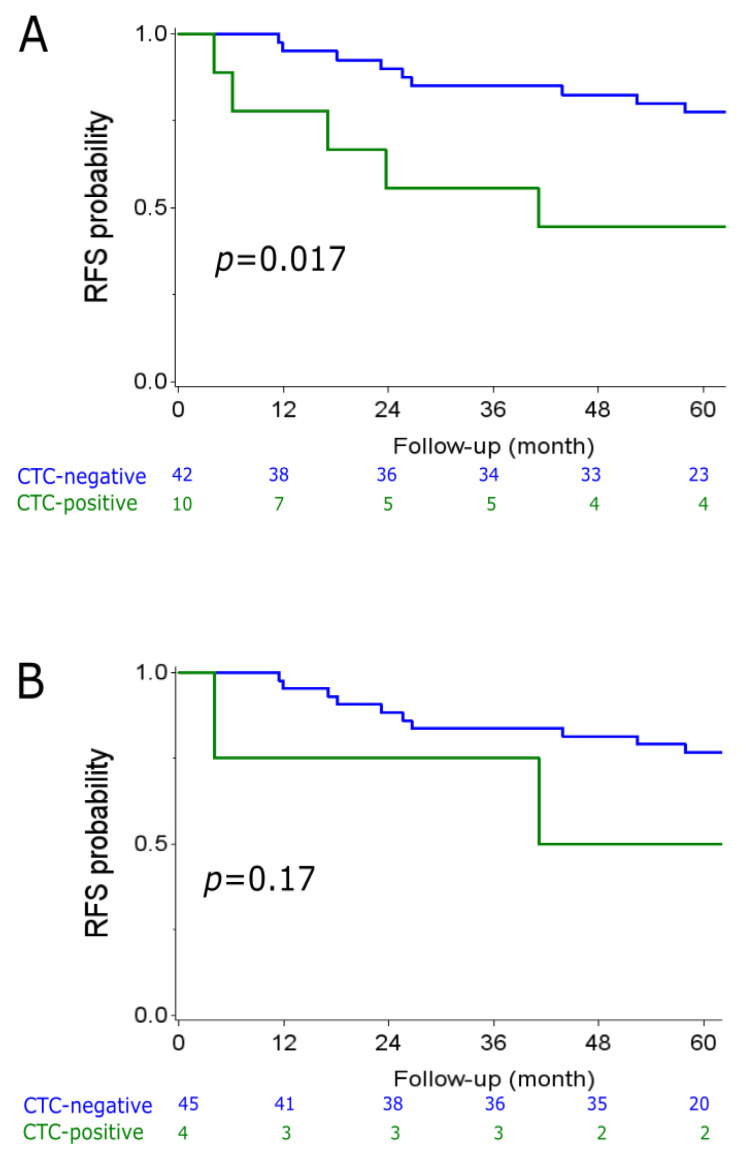
Kaplan–Meier curves for relapse-free survival (RFS) depending on the CTC status (green = CTC-positive; blue = CTC-negative). (**A**): CTC status at baseline. (**B**) CTC status after one cycle of treatment. *p* = Log-rank *p*-value.

**Figure 3 cancers-13-00140-f003:**
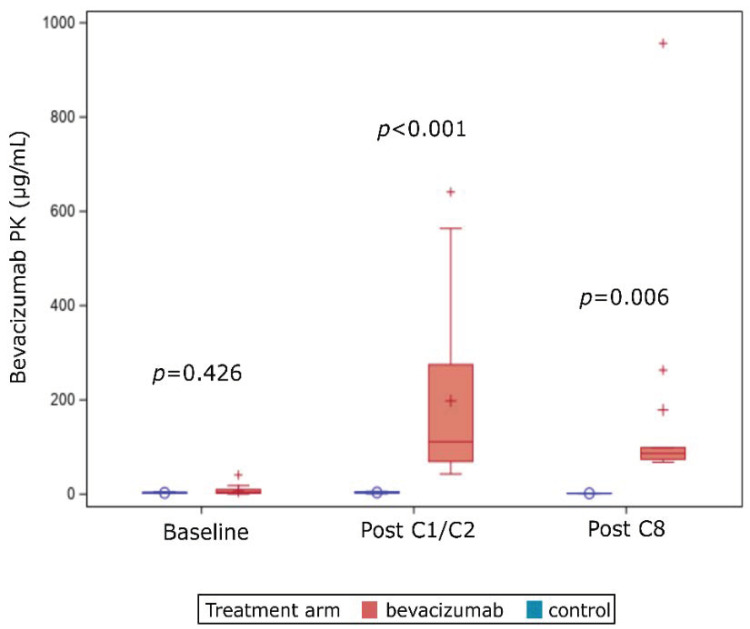
Box plots representing bevacizumab PK values at baseline, post C1-C2, and after treatment completion in both treatment arms.

**Figure 4 cancers-13-00140-f004:**
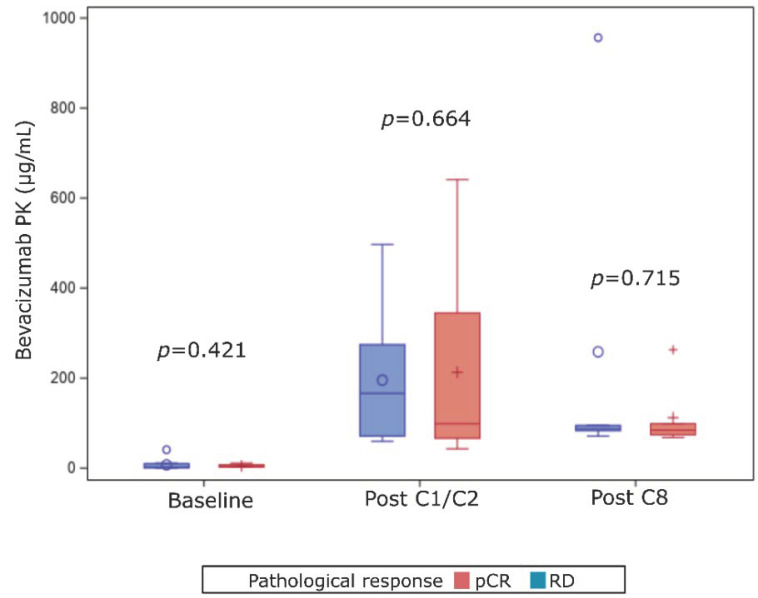
Bevacizumab PK values according to pathological response for patients enrolled in the experimental arm. pCR: pathological complete response; RD: residual disease.

**Figure 5 cancers-13-00140-f005:**
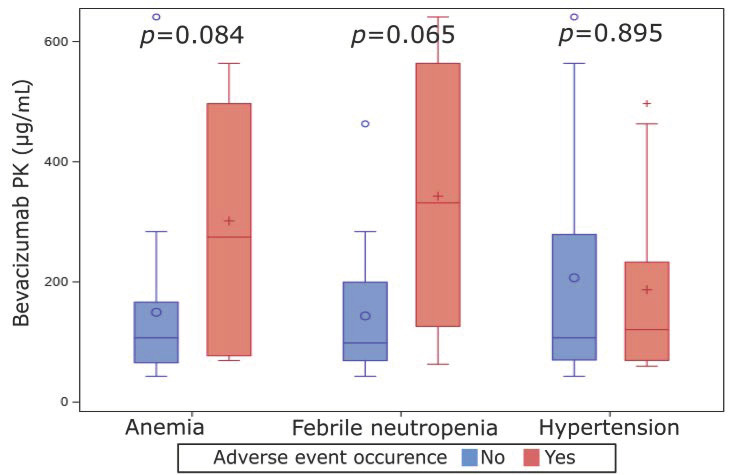
Correlation of post C1/C2 bevacizumab pharmacokinetics data to treatment-related adverse events for patients enrolled in the experimental arm of the AVASTEM trial.

**Table 1 cancers-13-00140-t001:** Correlation between circulating tumor cells (CTC) and demographics features. Data are expressed as N (%) unless otherwise specified.

Clinical Features	Baseline		*p*-Value	Post-C1		*p*-Value
	CTC-Negative (*N* = 45)	CTC-Positive (*N* = 15)		CTC-Negative (*N* = 51)	CTC-Positive (*N* = 5)	
Age, years (median, min–max)	50.4 (24.3–68.5)	56.6 (28.3–66.3)	0.83	50.6 (24.3–68.5)	48.7 (28.4–62.3)	0.46
Menopausal	9 (20)	4 (27)	0.72	10 (20)	1 (20)	1
Tumor size T2	20 (44)	3 (20)	0.16	21 (41)	1 (20)	0.61
Tumor size T3	14 (31)	5 (33)		17 (33)	2 (40)	
Tumor size T4	11 (24)	7 (47)		13 (25)	2 (40)	
Positive axillary lymph node	33 (75)	15 (100)	0.05	39 (78)	5 (100)	0.57
De novo metastatic	3 (7)	5 (33)	0.02	6 (12)	1 (20)	0.50

**Table 2 cancers-13-00140-t002:** CTC status at baseline and after one cycle of treatment. Data are expressed as N (%); *: rank-Wilcoxon’s test.

CTC Status		Bevacizumab Arm	Control Arm	*p*-Value *
CTC-baseline	negative	29 (76.32)	16 (72.73)	0.77
	positive	9 (23.68)	6 (27.27)	
CTC-post C1	negative	33 (94.29)	18 (85.71)	0.35
	positive	2 (5.71)	3 (14.29)	

**Table 3 cancers-13-00140-t003:** Correlation between CTC and pathological complete response. Data are expressed as N (%). pCR: pathological complete response, RD: residual disease, *: Fisher’s exact tests.

Pathological Response	CTC-Baseline		CTC-Post C1	
	positive	negative	positive	negative
pCR	3 (23.08)	13 (30.23)	1 (20)	14 (28.57)
RD	10 (76.92)	30 (69.77)	4 (80)	35 (71.42)
*p*-value *	0.74		1	

## Data Availability

Data presented in this study are available on request from the corresponding author.

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
