# Peer review of "Circulating Tumor Cells and Bevacizumab Pharmacokinetics during Neoadjuvant Treatment Combining Chemotherapy and Bevacizumab for Early Breast Cancer: Ancillary Analysis of the AVASTEM Trial"

_cancers, 2021, doi:10.3390/cancers13010140_

Round 1

Reviewer 1 Report

Sabatier and colleagues presented their findings by association of circulating tumor cells (CTCs) and bevacizumab pharmacokinetics with overall survival (OS) and recurrence-free survival (RFS) in patients who received bevacizumab in combination with chemotherapy and chemotherapy alone at univariate level. OS at 5 years was 54% (95% CI, 25-76) in CTC-positive patients compared with 91% (95% CI, 77-96) in CTC-negative patients (HR, 6.21, 95% CI, 1.75-22.06; P=0.005). RFS at 5 years was 44% (95% CI, 14-72) in CTC-positive cases compared with 78% (95% CI, 61-88) in CTC-negative cases (HR 3.51, 95% CI, 1.17-10.52; P=0.025). However, CTC status after one cycle of treatment was not significantly associated with OS nor RFS. In addition, serum bevacizumab levels were not associated with pCR nor OS nor the treatment-associated toxicities. The presentation of this manuscript is so confusing and lacks clarity at times.

  1. The CTC count at baseline as a prognosis factor in association with OS and RFS is consistent with other studies and the meta-analysis in the neoadjuvant therapy setting in breast cancer (Bidard et al). However, CTC count after one cycle of treatment was not significantly associated with the outcomes. The sample sizes were comparable at baseline and after one cycle of treatment (60 and 56 patients for OS; 52 and 49 for RFS). Discussion didn’t provide sufficient interpretations on their findings on lines 152-153. One cycle of therapy may not be adequate to impact durable outcomes. Please provide sound explanations according to recent progress on the treatment-associated prognostic factors - performance of the prognostic factors was driven by treatment modality (Nguyen et al., Association of independent prognostic factors and treatment modality with survival and recurrence outcomes in breast cancer. JAMA Network OPEN. 2020;3(7):e207213. doi:1001/jamanetworkopen.2020.7213
  2. It is unclear in this CTC and PK analysis cohort that how many patients were with nonmetastatic and how many with de novo metastatic breast cancer. In the context, it should provide a table on the demographics and tumor characteristics of the CTC and PK cohort or both side by side rather than the original trial. Are the CTC counts correlated with these demographic and clinical factors at baseline and after one cycle of therapy?
  3. Please indicate which variables were categorical and which variables were quantitative in the statistical analysis section.
  4. Wording in many places throughout the manuscript are causing confusions such as the term CTC clearance; line 100, remove “improvement”; line 150, switch ‘improvement of OS” to better OS; line 188-189, rewording as …nor to long-term survival in our dataset.

Author Response

Dear Editor,

Please, find enclosed a revised version of our manuscript cancers-1034913 by Sabatier and colleagues entitled “Circulating tumor cells and bevacizumab pharmacokinetics during neoadjuvant treatment combining chemotherapy and bevacizumab for early breast cancer: ancillary analysis of the AVASTEM trial” submitted for publication to Cancers

We thank the reviewers and the Editor for their positive and helpful comments, which have been taken into account as follows. As you will see, we have answered the questions raised by the reviewers and modified the manuscript as suggested.

Referee’s Comments to the Author and Authors’ Responses

Reviewer 1

Sabatier and colleagues presented their findings by association of circulating tumor cells (CTCs) and bevacizumab pharmacokinetics with overall survival (OS) and recurrence-free survival (RFS) in patients who received bevacizumab in combination with chemotherapy and chemotherapy alone at univariate level. OS at 5 years was 54% (95% CI, 25-76) in CTC-positive patients compared with 91% (95% CI, 77-96) in CTC-negative patients (HR, 6.21, 95% CI, 1.75-22.06; P=0.005). RFS at 5 years was 44% (95% CI, 14-72) in CTC-positive cases compared with 78% (95% CI, 61-88) in CTC-negative cases (HR 3.51, 95% CI, 1.17-10.52; P=0.025). However, CTC status after one cycle of treatment was not significantly associated with OS nor RFS. In addition, serum bevacizumab levels were not associated with pCR nor OS nor the treatment-associated toxicities. The presentation of this manuscript is so confusing and lacks clarity at times.

We first want to thank the Reviewer 1 for their very helpful comments that will surely help us to improve our manuscript. 

  1. The CTC count at baseline as a prognosis factor in association with OS and RFS is consistent with other studies and the meta-analysis in the neoadjuvant therapy setting in breast cancer (Bidard et al). However, CTC count after one cycle of treatment was not significantly associated with the outcomes. The sample sizes were comparable at baseline and after one cycle of treatment (60 and 56 patients for OS; 52 and 49 for RFS). Discussion didn’t provide sufficient interpretations on their findings on lines 152-153. One cycle of therapy may not be adequate to impact durable outcomes. Please provide sound explanations according to recent progress on the treatment-associated prognostic factors - performance of the prognostic factors was driven by treatment modality (Nguyen et al., Association of independent prognostic factors and treatment modality with survival and recurrence outcomes in breast cancer. JAMA Network OPEN. 2020;3(7):e207213. doi:1001/jamanetworkopen.2020.7213

We thank the reviewer for this very constructive comment. The following sentences have been added in the discussion section: “The low number of patients who were still CTC-positive after the first cycle of treatment may be an explanation of this absence of correlation. However, we cannot rule out that CTC measurement after one cycle of therapy is not an adequate surrogate marker of long-term efficacy and may not affect survival. Moreover, treatment modality (here bevacizumab vs no bevacizumab) may also have modified the prognostic value of CTC status. It has been recently shown that treatment modalities can indeed influence the prognostic impact of clinical and biological features [Nguyen et al. JAMA Network Open 2020].” 

  1. It is unclear in this CTC and PK analysis cohort that how many patients were with nonmetastatic and how many with de novo metastatic breast cancer. In the context, it should provide a table on the demographics and tumor characteristics of the CTC and PK cohort or both side by side rather than the original trial. Are the CTC counts correlated with these demographic and clinical factors at baseline and after one cycle of therapy?

We agree with the reviewer that the details of demographics according to the CTC and PK cohorts will make the reading and understanding easier. We thus have completed in the table S1 the details for each cohort.

Moreover, a new table (new Table 1) has been added including the correlation between CTC status (at baseline and post-C1) and clinical features. We also have completed the  first paragraph of the results section with the following sentences: “Among patients with CTC data available, percentages of positive axillary lymph node and De novo metastatic patients were significantly different between CTC positive and negative subgroups (100% vs 75% p=0.05 and 33% vs 7% p=0.02, respectively, Table 1). CTCs data were still available after one cycle of treatment for 56 patients (35 in the experimental arm and 21 in the control arm) with no significant differences between subgroups for demographic and clinical factors

  1. Please indicate which variables were categorical and which variables were quantitative in the statistical analysis section.

The corresponding sentence has been modified as follows in the Methods section: “Categorical variables (treatment groups, menopausal status, tumor stage, axillary lymph node involvement, treatment response and CTC status) were described using counts and frequencies, and quantitative variables (age and PK values) were described using medians and ranges”.

  1. Wording in many places throughout the manuscript are causing confusions such as the term CTC clearance; line 100, remove “improvement”; line 150, switch ‘improvement of OS” to better OS; line 188-189, rewording as …nor to long-term survival in our dataset.

The term “CTC clearance” has been deleted through the manuscript to improve the wording as recommended. We have also changed “improvement of OS“ to “better OS” in line 150. The sentence in lines 188-189 has also been modified and the whole manuscript has been rewieved by an English-native colleague.

Reviewer 2

Dear authors,

Please find the following comments:

We would like to thank Reviewer 2 for their constructive comments that helped us to improve our manuscript.

Abstract

Explain what is the difference in survival: is it OS, PFS? What is the difference in months of worse survival having CTC and being treated from the study? It should be clear from the abstract.

To make this part of the abstract easier to understand, we have modified it as follows, including details of the 5y-os and RFS: “The absence of CTC at inclusion was correlated to better outcome. Five-years OS was 91% for CTC-negative patients vs 54% for CTC-positive cases (HR=6.21; 95%CI [1.75-22.06], p=0.001, log-rank test). Similar results were observed for RFS with 5y-RFS of 78% vs 44% (HR=3.51; 95%CI [1.17-10.52], p=0.017, log-rank test).

Introduction:

CTC are 10-80% expressed in cancer cells. The number is inaccurate, not updated (the paper is of 2011) and based on only 1 study. To improve it please check the review “https://www.ecronicon.com/ecgy/pdf/ECGY-07-00217.pdf “ as a starting point to find more information on a larger and more updated variety of estimates based on much more updates studies. That short paragraph of the review could be improved.

This paragraph has been modified and replaced by the following sentences: “Circulating tumor cells (CTC) are released by the main tumor mass and may be at the origin of metastases spreading (Giuliano et al. 2014). They have been shown to be correlated to survival and to prediction of chemotherapy efficacy in the metastatic setting (Nakamura et al. 2010). CTC can be identified for 2 to 55% of patients with primary breast cancer, with a detection rate below 25% with the CellSearch® system which is the only assay that has received FDA approval (Thery et al. 2019). CTC detection in patients with early breast cancer is associated with tumor size, lymph node involvement, and high grade (Wülfing et al. 2006; Janni et al. 2016). CTC detection at diagnosis is also predictive of a poorer outcome with decreased disease-free survival and overall survival in chemonaive patients diagnosed with early breast cancer (Lucci et al. 2012; Riethdorf et al. 2017). Moreover, CTC detection after neoadjuvant chemotherapy is associated with worst survival for early triple-negative breast cancer with residual disease after preoperative chemotherapy (Radovich et al. 2020). No data exist concerning CTC value to predict bevacizumab efficacy for early non-inflammatory breast cancer. »

Results:

Make a table for the clinic-pathological features to summarize them. It would help the readers to better visualize how the data were collected from patients.

The table S1 has been completed with detailed data for both the CTC and PK cohorts.

Explain clearly what is the statistical difference in a more structure way. Explain Figure 1a and Figure 1b individually in the main text. For example: Figure 1A shows that 5yr-OS was statistically higher in patients expressing low levels of CTC (CI) compared to patients expressing high levels of CTC   (CI) (HR, CI; p=0.005). Do the same for figure 2A and 2B.

Figures 1 and 2 description has been modified as suggested. For example, the following paragraph has been added: “The absence of CTC at inclusion was associated with better outcome. Figure 1A shows that 5y-OS was statistically higher in patients without CTC at baseline (91% (95%CI [77-96])) compared to CTC-positive patients (54% (95%CI [25-76]), (Hazard ratio = 6.21 (95%CI [1.75-22.06]; p=0.001, log-rank test). This survival gain was not observed when analyzing CTC status after one cycle of treatment. Figure 1B shows that 5y-OS was similar for patients without CTC at C2 (86% (95%CI [72-93])) and for patients who were still CTC-positive at C2 (80% (95%CI [20-97])), (HR = 1.56; 95%CI [0.19-12.67]; p=0.68, log-rank test).”

Moreover, we have added the National Clinical Trail number (NCT01190345) in the introduction section and have standardized the p-values through the manuscript by keeping log-rank test p-values instead of Wald test p-values.

We hope that this improved version will meet with your approval for publication in Cancers.

Sincerely yours,

Dr Renaud SABATIER

Reviewer 2 Report

Dear authors,

Please find the following comments:

Abstract

Explain what is the difference in survival: is it OS, PFS? What is the difference in months of worse survival having CTC and being treated from the study? It should be clear from the abstract.

Introduction:

CTC are 10-80% expressed in cancer cells. The number is inaccurate, not updated (the paper is of 2011) and based on only 1 study. To improve it please check the review “https://www.ecronicon.com/ecgy/pdf/ECGY-07-00217.pdf “ as a starting point to find more information on a larger and more updated variety of estimates based on much more updates studies. That short paragraph of the review could be improved.

Results:

Make a table for the clinic-pathological features to summarize them. It would help the readers to better visualize how the data were collected from patients.

Explain clearly what is the statistical difference in a more structure way. Explain Figure 1a and Figure 1b individually in the main text. For example: Figure 1A shows that 5yr-OS was statistically higher in patients expressing low levels of CTC (CI) compared to patients expressing high levels of CTC   (CI) (HR, CI; p=0.005). Do the same for figure 2A and 2B.

Author Response

(The authors gave the same response as above.)

Round 2

Reviewer 1 Report

The authors have addressed my questions on the first round of their revision. However, the following points should be addressed.

Except addressing CDC in relation to pathological response, it should avoid use of the words “predict” or “predictive of” for reports of observational studies assessing prognostic parameters, unless your study design is a maker analysis such as CDC in a randomized manner or a diagnostic/prognostic study that has used an appropriate approach to quantifying predictive performance by, but not limited to, split sampling or cross-validation.

In Simple Summary: line 33, rewording the sentence as the bevacizumab concentrations in association with hematological toxicities after one or two cycles of treatment only reached borderline significance or have trends toward significance.

Abstract: Consider removing “CellSearch technology”, “ELISA” from the abstract. The CTC and PK analyses were only available for maximal 60 patients in this ancillary analysis and please delete “Seventy-five patients were included. Out of them 50 received bevacizumab-chemotherapy and 25 received chemotherapy alone”; Delete “For CTC-positive patients at baseline” on line 49. In the CDC cohort, it did include patients with de novo metastatic breast cancer (stage IV) for OS and was not appropriate just mentioning “stage II and III”; re-word the concluding sentence lines 52 to 54.

Please unify the use of CTC sampling timepoints either “after cycle 1” or “at cycle 2” throughout the manuscript.

Author Response

Dear Editor,

Please, find enclosed a second revised version of our manuscript cancers-1034913 by Sabatier and colleagues entitled “Circulating tumor cells and bevacizumab pharmacokinetics during neoadjuvant treatment combining chemotherapy and bevacizumab for early breast cancer: ancillary analysis of the AVASTEM trial” submitted for publication to Cancers

We thank the reviewer for their helpful comments, which have been taken into account as follows. As you will see, we have answered the questions raised by the reviewer and modified the manuscript as suggested.

Referee’s Comments to the Author and Authors’ Responses

Reviewer 1

The authors have addressed my questions on the first round of their revision. However, the following points should be addressed.

Except addressing CDC in relation to pathological response, it should avoid use of the words “predict” or “predictive of” for reports of observational studies assessing prognostic parameters, unless your study design is a maker analysis such as CDC in a randomized manner or a diagnostic/prognostic study that has used an appropriate approach to quantifying predictive performance by, but not limited to, split sampling or cross-validation.

We agree that the predictive value of biomarkers should be limited to tumor response assessment, and thus to pCR evaluation in our study. We thus have deleted the words “predict/predictive” when they were not used for pCR assessment.

In Simple Summary: line 33, rewording the sentence as the bevacizumab concentrations in association with hematological toxicities after one or two cycles of treatment only reached borderline significance or have trends toward significance.

We agree with the reviewer that to be more consistent with the p-values, we modified this sentence as follows: “However, bevacizumab concentration was tended to be associated with an increase of hematological toxicities during the study”.

Abstract: Consider removing “CellSearch technology”, “ELISA” from the abstract.

We think that mentioning the technologies used to perform these experiments are useful for the reader. We prefer not to remove them as they bring important details concerning the methods.

The CTC and PK analyses were only available for maximal 60 patients in this ancillary analysis and please delete “Seventy-five patients were included. Out of them 50 received bevacizumab-chemotherapy and 25 received chemotherapy alone”;

It is right that CTC analyzes were available for 60 and PK for 29 patients (experimental arm only). However, the 60 “CTC patients” did not include all 29 “PK cases”. Ancillary data were thus available for more than 60 patients. To be clearer we added the following sentence in the abstract: “CTC results were available for 60 patients and PK data for 29 patients in the experimental arm”.

Delete “For CTC-positive patients at baseline” on line 49.

This sentence has been deleted.

In the CDC cohort, it did include patients with de novo metastatic breast cancer (stage IV) for OS and was not appropriate just mentioning “stage II and III”; re-word the concluding sentence lines 52 to 54.

The mention of tumor stage has been deleted. The new concluding sentence is now: “In conclusion, CTCs detection at baseline is a prognostic marker for breast cancer receiving a neoadjuvant chemotherapy-bevacizumab combination independently of tumor response”.

Please unify the use of CTC sampling timepoints either “after cycle 1” or “at cycle 2” throughout the manuscript.

As suggested by the reviewer, the term “Cycle 2 time point” has been changed to “post-C1 time point” to avoid any confusion.

We hope that this improved version will meet with your approval for publication in Cancers.

Sincerely yours,

Dr Renaud SABATIER
